# Future Industrial Applications: Exploring LPWAN-Driven IoT Protocols

**DOI:** 10.3390/s24082509

**Published:** 2024-04-14

**Authors:** Mahbubul Islam, Hossain Md. Mubashshir Jamil, Samiul Ahsan Pranto, Rupak Kumar Das, Al Amin, Arshia Khan

**Affiliations:** 1Department of Computer Science, United International University, Dhaka 1212, Bangladesh; mislam201011@mscse.uiu.ac.bd; 2Department of Electrical and Electronic Engineering, Islamic University of Technology, Gazipur 1704, Bangladesh; mubashshirjamil99@gmail.com (H.M.M.J.); samiulahsan@iut-dhaka.edu (S.A.P.); 3College of Information Sciences and Technology, Pennsylvania State University—University Park, University Park, PA 16802, USA; rjd6099@psu.edu; 4Department of Information Systems, University of Maryland—Baltimore, Baltimore, MD 21201, USA; salamin1@umbc.edu; 5Department of Computer Science, University of Minnesota—Duluth, Duluth, MN 55812, USA

**Keywords:** Internet of Things (IoT), LPWAN, LoRa, Sigfox, Z-Wave, NB-IoT, LTE-M, RedCap

## Abstract

The Internet of Things (IoT) will bring about the next industrial revolution in Industry 4.0. The communication aspect of IoT devices is one of the most critical factors in choosing the device that is suitable for use. Thus far, the IoT physical layer communication challenges have been met with various communications protocols that provide varying strengths and weaknesses. This paper summarizes the network architectures of some of the most popular IoT wireless communications protocols. It also presents a comparative analysis of some of the critical features, including power consumption, coverage, data rate, security, cost, and quality of service (QoS). This comparative study shows that low-power wide area network (LPWAN)-based IoT protocols (LoRa, Sigfox, NB-IoT, LTE-M) are more suitable for future industrial applications because of their energy efficiency, high coverage, and cost efficiency. In addition, the study also presents an Industrial Internet of Things (IIoT) application perspective on the suitability of LPWAN protocols in a particular scenario and addresses some open issues that need to be researched. Thus, this study can assist in deciding the most suitable IoT communication protocol for an industrial and production field.

## 1. Introduction

Due to the recent emergence of 5G in the wireless telecommunications domain, the Internet of Things (IoT) is taking flight in many aspects of our day-to-day lives [1]. Moreover, the advancement of Industry 4.0 has brought digitization to the manufacturing sector. Industry 4.0, also called the Fourth Industrial Revolution, represents the realization of the digital revolution in the industry. It shifts the production procedure from a centralized control to a decentralized process [2]. It offers real-time decision making, increased productivity, flexibility, and agility to transform the ways businesses produce, develop, and market their goods [3]. In Industry 4.0, IoT devices need to be able to work at long ranges and during locomotion. Indeed, connecting numerous such devices under the same network is easier when performed on a wireless medium [4]. By 2028, IoT connections have the possibility to increase by around 21.5 billion, with a comparison shown between the number of devices in 2022 and a forecast for 2028 in [5]. IoT will mostly be used in smart parking and vehicle-to-vehicle communication, augmented maps, data collection, smart water supply, and smart appliances at homes and offices. Smart power grids, agriculture, and health sectors are also going to be reliant on the interconnectivity of IoT devices [6]. The deployment of IoT devices has some requirements that differ from conventional wireless telecommunication requirements. For example, a huge factor of IoT systems is scalability from a number of devices’ points of view. One of its characteristics is the massive number of devices it incorporates in a network, and it performs upon frugal power consumption by inexpensive devices [7]. According to Figure 1 [5], we have found that current trends and future trends both show that wide area protocols and short-range protocols-supported devices are mainly used while cellular protocols-supported devices are comparatively less in use. Considerable research [8,9,10] has supported the idea that for broad areas and long ranges, primarily industrial, scientific, and medical (ISM) band-based LPWAN protocols are superior to cellular IoT protocols. Three of the key requirements of IoT technologies are low power consumption and long-range communication at low cost [11,12]. In addition to these, parameters like data rate, security, link budget, and others should also be taken into consideration [13]. Different technologies occupying IoT telecommunications fulfill these requirements differently. Some of them offer a better range, while others offer lower power consumption. In order to decide which technology should be the most suitable in a certain deployment environment, it is essential to place these parameters side by side and obtain a clear view. Some of the most popular IoT communication protocols used today are Z-Wave, LoRa, the Narrow Band Internet of Things (NB-IoT), Sigfox, Long Term Evolution for Machines (LTE-M), and the 5G New Radio (NR)-based new protocol named Reduced Capability (RedCap).

On one hand, there are IoT protocols like Lora, Sigfox, and Z-Wave that operate in the unlicensed ISM spectrum [14]. LoRa is a deep-rooted exclusive framework in the IoT business that utilizes the Macintosh-level convention LoRaWAN and offers long reach and an extraordinary geo-inclusion [15]. Sigfox is one of the least power-consuming IoT frameworks right now [16]. On the other hand, systems like NB-IoT, LTE-M, and RedCap make use of the licensed frequency spectrum [17,18], resulting in higher power consumption but better quality of service. NB-IoT is a high-quality IoT technology that operates on a licensed spectrum and was made available in the 3rd Generation Partnership Project (3GPP) Release 13 [19]. It can support a large number of devices in a single cell, has a larger coverage area, has low device complexity, and offers flexibility in deployment [20]. Another LTE-based protocol for machine-to-machine (M2M) communication, LTE-M, was also included in 3GPP Release 13 and operates at the licensed frequency bands [21]. 3GPP Release 17 introduced RedCap devices, a new protocol based on the NR architecture that is suited for devices with lesser complexity, such as IoT devices [22].

There have been numerous approaches in order to present a comparison among the different IoT protocols. Some surveys present a comparison focusing on various IoT-based wireless sensor networks (WSN) but lack real-time testing [23]. Studies in [24,25] show a comparison among the devices regarding various parameters, albeit in a limited environment. All these surveys focus on various metrics and face distinct challenges, but to our best knowledge, not much attention has been given to three essential comparison matrices simultaneously, power, range, and cost, and the identification of the most suitable IoT protocol for future industrial use. The contributions of this article are the following:We compare and contrast the six most widely used IoT communication protocol standards (Sigfox, LoRa, Z-Wave, NB-IoT, LTE-M, and RedCap). In order to comprehend the fundamental differences between the technologies, we first provide a summary of their network architectures. Next, we compare their various important performance parameters, including power utilization, cost, range, data rate, QoS, and security. We also review them from an application perspective in various industrial settings.As per our knowledge, we are the first to make a comparative analysis of the NR-based protocol RedCap with ISM-based protocols.We, therefore, demonstrate that LPWAN protocols are the protocols of the future for the upcoming industrial revolution and offer the optimal application perspective and case studies for the Industrial Internet of Things (IIoT).Finally, we discuss some of the key challenges present in the current technologies that require attention and are open for future research exploration in IoT correspondence.

The rest of the article is illustrated as follows: Section 2 provides a brief summary and contribution to a comparative analysis of the existing literature. Section 3 describes available wireless IoT protocols. Section 4 contains a performance comparison of the targeted IoT protocols. Section 5 discusses the industrial application perspective of the relevant IoT protocols and summarizes them. Section 6 involves the scope for future works described in the surveyed literature. Finally, Section 7 brings a conclusion to the survey carried out in this research.

## 2. Literature Survey

In this section, we provide a brief overview and synthesis of significant research, most of which is based on comparative analysis among the IoT physical layer protocols based on technology, application, and performance.

In order to determine the optimal communication protocol, Andre Gl’oria et al. [24] thoroughly studied the key protocols currently in use, conducted a comparative analysis, and then selected protocols based on the findings. According to their findings, LoRa is a more trustworthy option for a wireless protocol since it requires little complexity and expense. Shadi Al-Sarawi et al. [25] overviewed IoT communication protocols’ visions, advantages and disadvantages, and additional QoS like energy consumption range and data rate to compare IoT communication protocols.

Ala’ Khalifeh et al. [23] evaluated several wireless technologies (LoRaWAN, NB-IoT, Sigfox, and LTE-M) to consider how they might be employed in fifth-generation (5G) communication technologies and wireless sensor networks. According to Thays Moraes et al.’s [26] studies, throughput, message size, and packet loss were used to evaluate how the Advanced Message Queuing Protocol (AMQP), MQ Telemetry Transport (MQTT), and Constrained Application Protocol (CoAP) behaved concerning speed and fault injection. According to tests, the researchers suggested that the CoAP protocol is an intriguing option for applications with limited network resources. A concentrated study by Mroue et al. [27] described the characteristics of the medium access control (MAC) layer for the low-power wide area network (LPWAN) options that are currently available, such as LoRa, Sigfox, and NB-IoT. The displaying of a thick organization for every one of these innovations was likewise canvassed in this review. The characteristics of various systems, such as carrier frequency, packet length, channel count, and spectrum access, were compared in this research contribution. The scientists made a model framework with NB-IoT gadgets, an IoT cloud stage, an application server, and a client application to show the benefits of NB-IoT when joined with other LPWA innovations in [28].

Researchers in [29] introduced a concise examination of the superior organization limit, gadget life span, and cost of LoRaWAN and Sigfox. NB-IoT, then again, performed better in terms of inactivity and administration quality. Moreover, they examined the different application situations and the innovation that is best in helping future scholars and business experts. Mikhaylov and others showed the numerous options that LoRaWAN geographies portray and demonstrated a wide range of situations considering the actual layer and zeroing in on the issue of organization security [30]. Adelantado et al. [31] illustrated the features and drawbacks of the LoRaWAN protocol, which were explored when a strategy was taken to decide the use case of LoRaWAN technology; furthermore, in which use cases it does not function. In their review, Rashmi et al. [32] analyzed and depicted the mechanical contrasts between LoRa and NB-IoT with respect to the actual qualities of network engineering. The examination was introduced as a near and engaging review. Although the model that Hendrik et al. [33] provided is based on the physical layer, it does illustrate some intriguing insights into the decoding performance in LPWAN when there is packet collision. Nadège Varsier et al. analyzed the potential of RedCap, previously known as NR-Lite, in Enhanced Mobile Broadband (eMBB) and Ultra-Reliable Low Latency Communications (URLLC) use cases in contrast to NB-IoT and LTE-M which, are more suited for Massive Machine-Type Communications (mMTC) criteria. They compared the NR-based RedCap with the LTE-based NB-IoT and LTE-M from bandwidth, coverage, data rate, latency, reliability, and battery life points of view and focused on RedCap’s appropriateness for Industry 4.0: wearables, video surveillance, and IIoT [34]. Researchers in [35] presented a comparative review among only the cellular-IoT protocols LTE-M, NB-IoT, RedCap, and Extended Coverage GSM IoT (EC-GSM-IoT) with respect to bandwidth, data rate, coverage, mobility, device capacity per cell, modulation, and others. Table 1 presents a concise summary of the already existing relevant literature and offers a comparative point of view of the nature of the publications and the value they add to this research area.

Several studies [24,25,26] compared a few IoT protocols in an experimental setting, while many IoT protocols have been evaluated in simulated settings without an experimental setup. Additionally, some studies [23,29,34] conducted a theoretical analysis of IoT wireless communication protocols. In contrast to earlier efforts, our study suggests a comparative analysis based on architecture, performance evaluation of specific chosen features of different IoT protocols, and future application of the most suitable protocols based on different industrial use cases. Range, cost, and energy-consumption-based evaluation are also included because those factors will be vital to future industrial IoT applications. In order to advance the field of research, we also highlight a few common challenges for wireless communication protocols in industrial applications.

## 3. Wireless Protocols Architecture

This section summarizes the characteristics and features of some notable wireless communication protocols that are used for IoT applications where low power and medium to long ranges are basic requirements.

### 3.1. LoRaWAN

LoRaWAN was developed by the LoRa alliance for applications requiring long-range and low-power wide area networks (WANs) [38]. It enables highly efficient WANs by reducing the necessity of repeaters between nodes, leveraging its very low power consumption. This increases the overall system efficiency by reducing the number of required devices and also increasing the battery life of the nodes. LoraWAN utilizes the Sub-Ghz ISM band [39]. A noteworthy drawback of LoRaWAN is that the data rate is low [40], thus catering to only some specific IoT applications. The ultra-low power consumption also means that only star topology can be implemented, where all the nodes send information to the central gateway, which communicates with LoRa servers [41,42]. LoRa utilizes the chirp spread spectrum (CSS) modulation scheme [13]. A CSS transmit signal can be shown as follows:(1)$$\begin{array}{c} x_{k}\left[n\right]=\sqrt{\frac{E_{s}}{N}}exp\left(j\frac{2\pi }{N}kn\right)c\left[n\right] \end{array}$$

$E_{s}$ is the signal energy, $c\left[n\right]$ is the discrete time chirp signal with a period of *N*, and *k* is the data symbol:(2)$$k=\sum_{i=0}^{SF-1}2^{i}{\left[b\right]}_{i}$$
and *b* is the bit-word, as such:(3)$$b\in {\left\{0,1\right\}}^{SF}$$

The data rate of LoRa transmission, $R_{b}$, depends on the modulation bandwidth BW and the spreading factor SF:(4)$$R_{b}=SF*\frac{BW^{SF}}{2}$$

### 3.2. Sigfox

Sigfox is an open-source technology and the first of its kind. Sigfox was designed to focus on an even longer range and lower data rate compared to LoRaWAN, in which it utilizes ultra-narrow-band (UNB) frequency [43]. It was designed using a variety of low-power IoT devices, including several sensors and M2M applications [13]. It delegates the complex processing to the cloud instead of the individual nodes to reduce resource consumption [29]. Star topology can be implemented with Sigfox [41].

Figure 2 gives a simple overview of the network architecture of IoT technologies that utilize the free-of-cost ISM frequency spectrum to establish and maintain wireless M2M connectivity. In this framework, the data flow from the user end (UE) devices to the IoT base station over the associated wireless protocol: LoRa, Sigfox, or Z-Wave. For example, Sigfox uses its proprietary UNB technologies while utilizing the open-source sub-GHz ISM band spectrum. LoRaWAN also employs a sub-GHz ISM band spectrum for wireless connectivity with end devices. The end device can be an actuator, a temperature-measuring device, a sensor, etc. The base station then acts as the gateway to the respective protocol’s IoT server over an ethernet connection. Gateways have point-to-point connectivity with the servers through an IP network, and the data are passed on through the proprietary IoT cloud to the servers. Servers are placed in the network according to requirements, following the specifications supported by the topologies that are being used. Both Sigfox and LoRaWAN support star topology, while Z-Wave can implement mesh topology [41,42].

### 3.3. NB-IoT

NB-IoT is an IoT communication protocol that was designed to reduce some of the capabilities of legacy LTE while simultaneously enhancing the ones necessary for IoT networks [44]. This makes NB-IoT an appropriate choice for IoT applications, as it is suitable for low-power IoT devices. As IoT end devices do not need constant back-end broadcasting, the broadcasting frequency is reduced, which decreases the power consumption [29,44]. One has to keep in mind that this technology shares the same 3GPP licensed frequency bands with the LTE bands [45]. As a result, only certain operation modes are possible in order to ensure interference-free operation. NB-IoT uses the Quadrature Phase Shift Keying (QPSK) modulation scheme to modulate the bits [46]. A QPSK-modulated transmit signal can be shown as follows:(5)$$S_{QPSK}\left(t\right)=\frac{1}{\sqrt{2T}}a_{1}\left(t\right)cos\left(2\pi f_{c}t+\frac{\pi }{4}\right)+\frac{1}{\sqrt{2T}}a_{2}\left(t\right)sin\left(2\pi f_{c}t+\frac{\pi }{4}\right)$$

A binary bit stream of $a\left(t\right)$ with a period of *T* is demultiplexed into two different bitstreams $a_{1}\left(t\right)$ and $a_{2}\left(t\right)$, that represent +1 and −1 before multiplying by sine and cosine carriers to form a QPSK signal (Figure 3).

### 3.4. LTE-M

LTE-M is another protocol based on the traditional LTE [47]. However, it was designed to transmit bits at a higher data rate at the expense of comparatively higher power utilization and lower battery life compared to NB-IoT [48]. It was designed to cater to applications requiring relatively higher speeds and lower latency [13]. This protocol is useful for certain scenarios where the other protocols do not meet the data-rate requirements.

Figure 3 illustrates the network connectivity for 3GPP licensed band-based protocols such as NB-IoT and LTE-M. This architecture can be divided into two parts, which are the User Plane and Control Plane. The User Plane is concerned with the process of sending and receiving user data. The Control Plane manages the control processes that are implemented to establish communication and authentication of the end devices. The data are transmitted wirelessly from the UE IoT devices to the eNodeB by adopting the LTE-based IoT protocols. The bits are then transferred over a wired network to the Evolved Packet Core (EPC) network. The data are received by the Mobility Management Entity (MME) within the EPC, which focuses on eNodeB signaling, mobility, and security. The Home Subscriber Server (HSS) is associated with user authentication and profile. The Serving Gateway (SGW) handles the routing and packet forwarding towards the uplink and the downlink. The Packet Gateway (PGW) connects the packet core network with the external IP network. The data then flow through the IP network towards the IoT servers [49,50].

### 3.5. Z-Wave

Z-Wave is a relatively short-range communication protocol widely adopted for IoT applications, which was designed by Sigma Systems [51,52]. It can connect up to 50 devices in a smart home or small-scale commercial environment. Even though it operates in the ISM band, the details and specifications are not open source and can only be accessed by entities who have signed an agreement with Sigma Systems [53]. This protocol is geared towards medium-short range IoT networks where low power consumption for the nodes is a prerequisite, along with a low data rate. Mesh topology can be implemented for Z-Wave [25,54,55].

### 3.6. RedCap

5G consists of two radio technologies: the older LTE and the novel radio interface technology named New Radio (NR) [56]. While NB-IoT and LTE-M work on the modified LTE-based architecture, in Release 17 3GPP focused on the IoT use criteria by reducing the existing capabilities of the NR architecture to meet the requirements falling between more extreme requirements defined by mMTC, URLLC, and eMBB [22]. Thus, we were introduced to NR-Lite, which later became named RedCap. Some of the reasons to look into RedCap, although NB-IoT and LTE-M offer cellular IoT solutions, are better system efficiency than LTE due to beam-formed operation, higher subcarrier spacing for latency reduction, massive MIMO coverage, mixed numerology, higher positioning accuracy, and low-overhead carriers. Deployment in the Frequency Range 2 (FR2) spectrum can be very useful for private networks due to its limited range and high spatial reuse. Also, the 5G core and architecture (Figure 4) offer network slicing, service-based architecture, and flow-based QoS [57]. RedCap can become a successor to NB-IoT, offering lower latency and higher reliability due to its 5G backbone [58]. RedCap uses Quadrature Amplitude Modulation schemes such as 256 QAM and 64 QAM to modulate the bits [22,59].

## 4. Performance Comparison

This section provides an empirical guideline for selecting the appropriate communication protocol by comparing the IoT communication protocols. In order to compare them, parameters such as standard, energy consumption, coverage, data rate, security, modulation type, cost, and other factors are featured.

### 4.1. Data Rate

Data rate is crucial in IoT applications because it determines how quickly and efficiently devices can communicate, affecting real-time data analysis and system performance. High data rates support the seamless operation of interconnected devices, enabling faster decision making and better user experiences. Among the protocols, RedCap can provide the most significant data rate, up to 150 Mbps in downlink and 50 Mbps in uplink, in some use cases where QoS is essential [22,34]. LTE-M provides a data rate of up to 1 Mbps, while Sigfox, LoRa, Z-Wave, and NB-IoT have data rates of less than 1 Mbps [60,61,62]. Although Sigfox utilizes the UNB very efficiently and provides high receiver sensitivity and low-cost antenna due to very low-level noise, it falls short of the data rate, offering only 0.1 kHz [60]. In the case of NB-IoT, compared to the downlink rate, the uplink data rate is lower and up to 20 kbps [60]. On the other hand, LoRaWAN is perfect for long-distance transmission of small payloads, such as sensor data with small data rates [63].

### 4.2. Range

The range is essential in IoT applications because it defines the maximum distance devices can communicate, directly impacting the network’s scalability and deployment flexibility. A longer range allows for wider coverage and connectivity, especially in remote or large-scale environments. NB-IoT and LTE-M have far less coverage compared to the ISM band-based LPWAN protocols, with Sigfox having a maximum range of up to 40 km, LoRa having a good range of up to 20 km, and Z-Wave up to 30 m [60,61,62]. To place the matter into perspective, using Sigfox, we can cover a typical large city with just one base station. LoRa has a lower reach, and it can take up to three base stations to cover the entire city. The spreading factor, which determines how far and robustly the signal can travel, can increase LoRa’s transmission range [6]. However, it also increases the power consumption. Compared to LoRa, NB-IoT offers wider coverage in vehicle traction in agricultural applications [64]. NB-IoT and LTE-M primarily focus on applications where the end devices have issues with typical cellular networks or ISM band-based technologies due to physical barriers (indoor applications). In contrast, Z-wave focuses on even shorter-range indoor applications compared to NB-IoT and LTE-M [60]. On the other hand, RedCap offers coverage of less than 30 m, mainly suitable for indoor deployment [65].

### 4.3. Energy Consumption

IoT applications require careful consideration of energy consumption because many devices run on batteries and must last long before recharging to ensure long-term, sustainable operation. Low energy consumption, especially in isolated or difficult-to-reach locations, facilitates the effective deployment of large-scale IoT networks. LoRa, Sigfox, and Z-Wave are developed for portable devices with low battery capacity because of their low energy usage. They have minimal power usage as a result. In contrast, NB-IoT and LTE-M consume more energy than LoRa and Sigfox. Among these five protocols, LoRa has the highest energy efficiency of them all [61,66,67]. Although LTE-M and NB-IoT both have medium energy consumption rates, LTE-M performs better in favorable coverage conditions and improves as the conditions improve. In contrast, NB-IoT performs better in unfavorable coverage conditions [48]. However, for synchronous communication and QoS management, NB-IoT UE devices require more peak current, and its Orthogonal Frequency-division Multiplexing (OFDM) or Frequency Division Multiple Access (FDMA) access modes use more power [68]. RedCap’s energy consumption is higher than the other two cellular protocols: NB-IoT and LTE-M. It focuses more on lower latency and higher data rate compared to the others at the expense of higher power consumption [22].

### 4.4. Cost

Cost is an essential parameter for application development in large-scale technologies. Implementation, planning, and operation and maintenance periods are considered for cost assumption. Each of the periods has key requirements like frequency spectrum, energy efficiency, battery life, and device density [69,70]. In terms of cost, the ISM-band-based Sigfox, LoRa, and Z-Wave are all cheaper than the cellular-band-based NB-IoT, LTE-M, and RedCap. Among the cellular IoT protocols, RedCap is supposed to be more expensive than both NB-IoT and LTE-M [57]. Although Sigfox and LoRa are significantly cheaper than LTE-M and NB-IoT, respectively [25,47,71,72,73], between LoRa and Sigfox, LoRa is the more expensive one. LoRa has a long battery life by the framework, so the cost of replacing devices can be reduced. As architecture follows traditional wireless protocol regulation, deployment is straightforward. Similarly, Sigfox is designed to maximize energy efficiency by the ingrained framework. When it transmits data, Sigfox consumes very low power, and hardly any maintenance is required [73]. The above-stated features in the design help reduce the cost of LoRa and Sigfox. On the other hand, LTE-M uses power consumption features to extend device health for a long time, although licensed band standard and long coverage framework are more expensive than LPWAN protocol [25,73]. Finally, in terms of device density, LoRaWAN is cost-efficient where the device density is low, and in metropolitan regions with a high device density, NB-IoT and LTE-M are cost-effective [74].

### 4.5. Security

The inherent complexity of IoT architecture makes security and privacy issues extremely challenging. IoT systems’ security requirements involve confidentiality, integrity, authentication, authorization, access control, and availability. The scalability of existing security systems for authenticating and controlling access to enormous IoT resources has prompted the industry and researchers to pursue a decentralized approach. Recently, more lightweight and highly efficient encryption techniques have been developed to safeguard the smallest IIoT, such as edge devices, sensor nodes, and WSNs [75]. All the IoT communication protocols utilize authentication and encryption procedures in terms of security. Whereas Sigfox employs low-level authentication and encryption, LoRa, Z-Wave, NB-IoT, and LTE-M use the Advanced Encryption Standard (AES) [76] block cipher with counter mode. AES is safer than authentication and encryption. Compared to AES, authentication and encryption is extremely quick but exposed [73,77].

### 4.6. QoS

According to studies, cellular NB-IoT and LTE-M offer very low latency [78,79]. LoRaWAN, in contrast to Sigfox, offers lower bidirectional latency. Speed and low latency are significant benefits of LTE-M. LTE-M can deliver speeds of up to 1 Mbps in the uplink and 384 Kbps in the downlink. In addition, LTE-M has a 50–100 ms latency, its nodes can transmit at more effective data rates, and the reduced latency allows for real-time communication between the nodes [47]. For RedCap, latency can vary from less than 500 ms to 10 ms depending on its use case [22,34].

While comparing IoT communication protocols, Sigfox emerged as the future protocol because of its low cost and broad reach. However, a low-power consumption module will work best for LoRa. Overall, LoRa is more compatible with all environments because of its security features and low cost, great range, and low power consumption [29,47]. Table 2 provides a side-by-side juxtaposition of six dominant IoT communication protocols that are two wide-area protocols based on ISM bands (LoRa and Sigfox), three cellular protocols (NB-IoT, LTE-M, and RedCap), and a short-range ISM-based protocol (Z-wave). Figure 5 provides a graphical comparison among the different types of IoT communication protocols (LTE-based NB-IoT and LTE-M, NR-based RedCap, ISM-based short range Z-wave, and ISM-based wide area network Sigfox and LoRa) from coverage, data rate, and energy efficiency points of view. The figure shows that LoRa and Sigfox are the most energy efficient while also delivering the longest coverage. Their low power consumption, high coverage, and cost-effectiveness make them indispensable solutions for IIoT applications in diverse use cases. While Z-Wave and the cellular technologies can offer better data rates compared to them, their power consumption is higher.

Among all the requirements regarding IIoT communication systems, cost efficiency, coverage, and energy efficiency are three features of prime importance. The purchase and deployment of IoT devices need to be as inexpensive as possible because a large number of devices need to be deployed to create an efficient sensing or monitoring system in the industrial environment.

Also, IoT systems will often be used across small to large geolocations depending on the industry size and purpose that need to be covered by the system. That is why transmission range and coverage are very important factors when deciding the suitable protocol for a scenario. Lastly, in some cases, IoT devices work in such industrial systems where continuous transmission and reception of data is not always necessary, but longevity is essential. Therefore, energy efficiency carries great significance in this area of communication protocols. Figure 6 provides a graphical comparison among the six types of IoT communication protocols from coverage and cost-efficiency points of view. Proprietary protocols such as LoRa, Sigfox, and Z-Wave are considerably cheaper compared to the licensed spectrum-dependent cellular options. Among them, Sigfox and LoRa can provide a good range, which makes them the best candidate for IIoT use cases. In Figure 7, we present a graphical comparison of the six IoT communication protocols (LoRa, Sigfox, Z-Wave, NB-IoT, LTE-M, and RedCap) from the perspective of the five most important parameters: transmission range, energy consumption, cost, battery lifetime, and data rate. Their low power consumption, broad coverage, and cost-effectiveness make them indispensable solutions for IIoT applications in diverse use cases. The rest of this paper explores the industry-specific understanding of LPWAN-based IoT deployment in various industrial application areas.

## 5. Industrial Application Perspective

In order to evaluate the applicability of LPWAN and cellular IoT protocols and comprehend the benefits and constraints of the technology, a variety of LPWAN-based industrial applications are taken into account. Industrial Internet of Things (IIoT) applications have a comprehensive area of use [91]. This section discusses several industrial application use cases and provides an overview of the technologies that best fit each situation in terms of characteristics. Figure 8 illustrates the prime application area of LPWAN-based protocols in IIoT.

### 5.1. Smart City

Large-scale uses of IoT technology, such as intelligent parking, automated lighting, and smart garbage collection, are being developed [31,92] for Smart City applications. Similar to smart garbage collection systems, smart lighting systems respond to a measure with long variable periods by acting or reporting information [93]. Even though there is no significant dependency upon latency and jitter, in certain cases, the triggering factor is concurrent for many UE devices. LoRaWAN and Sigfox are acceptable solutions in this kind of scenario since they can cover large areas and a sizable number of user equipment at the cost of increased latency, collision, and jitter rates [25,31,40,71]. For example, Poddar et al. [94] conducted a case study regarding intelligent city applications in Estonia. They investigated the coverage analysis of two LPWAN technologies, NB-IoT and Sigfox, on university campuses in Estonia’s two major cities, Tartu and Tallinn. The results showed that Sigfox and NB-IoT give continuous coverage in outdoor areas, whereas NB-IoT performs better indoors.

### 5.2. Intelligent Logistics and Transportation

Logistics and transportation are two essential foundations of the anticipated IoT development over the coming years. Most applications aim to increase efficiency in sectors like cargo or public transportation. However, while some specific applications can tolerate jitter, delay, or unreliability, other applications cannot [31]. Due to the enormous quantity of data generated by sensors deployed on cars or roadside units (RSUs), intelligent transportation systems (ITSs) may have extra communication overhead, high bandwidth consumption, and more significant reaction delays while transmitting data to cloud servers. The cellular LPWAN architectures LTE-M and NB-IoT have been used by ITS applications to meet the challenges mentioned earlier. Also, LTE-M and NB-IoT are appropriate for providing backhaul infrastructure for ITS applications [95]. NB-IoT and LTE-M are the best suited for these applications because of their diversity, range, QoS, and low latency [95]. In particular, a smart parking management system was developed using NB-IoT to mitigate the high power consumption and high deployment costs of wireless networks. The proposed system has been deployed in two cities in Zhejiang Province, China, to effectively improve the utilization of existing parking facilities [96].

### 5.3. Smart Farming and Agriculture

Long battery lives are needed for sensor equipment in the agricultural sector. In a conventional farming environment, short coverage is sufficient. According to [29,97,98], utilizing temperature and humidity sensors in the irrigation industry might drastically cut water use while increasing productivity. Considering that a farming environment generally remains the same, devices only update sensor data a few times per hour. Sigfox and LoRaWAN are perfect for this sort of technology due to their low power consumption [29,99,100]. Codeluppi et al. [101] presented a low-cost, modular LoRaWAN-based IoT platform for improving the management of generic farms. The suggested platform was tested on farmland in Italy, gathering environmental data (air/soil temperature and humidity) relevant to the growth of agricultural goods (namely, grapes and greenhouse vegetables) over three months. According to the result, LoRaWAN performed optimally regarding data transmission and energy efficiency observed in indoor and outdoor areas. Also, many agricultural businesses still need legacy LTE-cellphone-based application connectivity due to financial and technological constraints. As a result, the NB-IoT cloud might not be a viable solution for agriculture in the future [29].

### 5.4. Smart Home

Property supervisors often use alerting measures such as temperature, humidity, safety, water flow, and electric plug sensors to prevent damage and promptly respond to requests without monitoring [102,103]. This is a normal use of a smart home framework. According to [29,104], these sensors need reasonably priced, long-lasting batteries. Sigfox and LoRaWAN are better suited for this type of application since they do not require frequent communication or high-quality service, and short-range operation is acceptable. For instance, Vatcharatiansakul et al. [105] evaluated the performance of LoRaWAN in a real-world environment in Bangkok, Thailand. Researchers concluded that communication ranges in both outdoor and interior settings are limited. As a result, IoT applications using LoRaWAN technology can be dependable in constrained communication ranges, such as the home or indoor environment.

### 5.5. Terminals for Retail Sales

Since they deal with regular contact, sale point systems demand guaranteed quality of service [29,106]. There is no limit on battery life because these devices have a constant electrical power source. Low latency is also crucial; otherwise, it limits the transactions a store can process within a given time [29]. As a result, NB-IoT and LTE-M are more appropriate for this application. Cost is also a consideration for retail point-of-sale terminals, making NB-IoT preferable over LTE-M.

### 5.6. Smart Environment

IoT-based innovative environments contain information about water quality, lowering levels of pollution in the air, lowering temperatures, preventing forest fires and landslides, tracking animals, monitoring snow levels, and early detection of earthquakes [13,107]. This type of project calls for sensors with long battery lives and also takes coverage and range into account. However, they also require high QoS, large bandwidth, and efficient bypassing of interference. Also, the projects are often carried out by large-scale undertakings that can make higher expenses bearable. As a result, NB-IoT and LTE-M are more appropriate for this sort of application [108]. However, the administration of smart water grids can benefit from the deployment of LoRaWAN [109]. The LoRa technology is also suitable for long-distance communication. Villarim et al. [110] conducted an experimental study in Portugal and Brazil to evaluate the LoRa communication range and received signal strength indicator (RSSI) in urban and natural settings. The results showed that LoRa is robust and suitable in dense urban environments, with a possible 2.1 km distance connection.

### 5.7. Energy Management

In order to build an industrial-level smart grid energy metering environment based on IoT, network control, load adjusting, remote observing and estimation, transformer wellbeing checking, and observation of wind plants and sunlight-based power establishments are a few significant variables [13,25]. Long range, low power, robust QoS, and excellent readability are requirements for high-level smart grid and energy metering. NB-IoT is, hence, better suited for usage in this application. Additionally, at the point where the power distribution section consists of individual commercial and residential appliances, LoRaWAN technology simplifies the transition to the private area network (PAN) and home area network (HAN) [111]. This includes smart electric bill meters, smart hazard alarm systems, etc. A case study was conducted in Lebanon to observe the performance and feasibility of deploying LoRaWAN for smart metering at a low cost and long range in a real-world scenario [112]. This solution offers an open-source, inexpensive, energy-smart metering system requiring little intervention in an already-existing electrical installation. In addition, the energy sector has been altered by LPWAN technology, and efficient sensor monitoring systems have reduced factory energy use [113]. As a result, the industrial energy system has become a major part of the IIoT. Furthermore, IIoT technologies have improved the efficiency of modern energy systems. For example, advanced control systems, predictive maintenance, and remote monitoring can improve smart energy management [114]. Software-defined machines, big data analytics, and smart sensors are emerging technologies that are steadily improving the system’s operating performance [115]. Smart power grids are also able to utilize RedCap’s low latency and high data rate IoT services [22].

### 5.8. Manufacturing and Automated Industries

LPWAN technologies play a vital role in developing cost-effective solutions in predictive industrial maintenance. Predictive maintenance involves the monitoring of the industrial equipment [116] and predicting the required maintenance [117] when necessary. It can reduce the possibility of unwanted shutdown of the equipment and reduce costs. LPWAN also provides solutions for effective and cost-efficient asset monitoring [13] due to its broad range of communication and minimal power consumption capabilities. LPWAN enables continuous and real-time tracing of the assets that help in optimizing asset utilization and improving overall resource allocation [118,119]. It also improves the supply chain visibility [10] by tracing and monitoring the shipment of goods and optimizing the supply chain operations. LPWAN is suitable for deployment in the factories as remote controls and industrial sensors to monitor processes (temperature, pressure, flow rate, asset health monitoring) and industrial environments. This will enable safer working environments in plants and factories [34]. Through a case study, Beliatis et al. [120] tried to identify a suitable technology for product traceability in the metal fabrication sector. The authors found Sigfox to be the most appropriate technology for tracking products during manufacturing due to its ability to transmit/receive data over a more extended range. On the other hand, RedCap is suitable for creating private networks in factories without interference, maintaining privacy and reliability [121].

### 5.9. Asset Tracking and Monitoring

IoT-based asset tracking and monitoring applications are becoming more common in the industry. Effective and scalable tracking and monitoring are essential for rental assets and industrial bases. Numerous tracking options are available, with uses spanning from monitoring automobiles and bicycles to monitoring company assets, including parcels, service goods, and containers. Technologies like LPWAN tracking are becoming quicker and more accurate [122,123] and, thus, can help the scalability of monitoring and tracking. LoRaWAN is being utilized as a geolocation tracker, where the differential time of arrival method determines the asset’s precise location at the closest gateway [124,125]. LoRaWAN and Sigfox are suitable options as they can handle an extensive coverage area and a significant user base at the expense of higher collision, latency, and jitter rates [122,124,126]. Significant research was performed to check the suitability of using LoRaWAN to conduct temperature studies on local government assets in the City of Melville, an Australian metropolitan area, and Curtin University’s residential areas. The results found that LoRaWAN is appropriate for this kind of application because of its long range and suitable data rate [127].

### 5.10. Wearables and Health

Using different IoT communication protocols, it is simple to observe and work on a patient’s health-related parameters, operate connected medical environments, healthcare wearables, patient surveillance, telemedicine, fall detection, athlete care, track chronic diseases, and track mosquito and other similar insect populations [13]. Most systems require low latency, diversity, range, and QoS, making NB-IoT a great fit for these applications. The healthcare business is tremendously benefiting from IIoT applications. They save money by allowing remote control of medical equipment, home-bound patient care, modeling, and monitoring [128]. As a result, hospitals benefit from innovative equipment that reduces patient wait times and enhances equipment performance. The growing popularity of mobile internet connections has accelerated the expansion of IIoT-powered in-home healthcare (IHH) services [114]. Patient-centered medical home (PCMH) care is a straightforward answer to numerous problems in the healthcare sector, such as chronic disease management, misuse of emergency departments, patient satisfaction, excessive medical expenses, and accessibility. Sensor devices provide significant information about patient health and aid in diagnosing disease [129]. Furthermore, the IIoT application domain provides telemedicine solutions, such as notifying patients about their wellbeing and monitoring their health with advanced medical equipment [130]. Healthcare wearables, patient monitoring, indoor remote health, PCMH care, and wellbeing monitoring are examples of applications that can be operated with shorter-range IoT protocols like Sigfox and LoRa [50,131]. A case study in Finland showed that LoRaWAN technology is suitable for wellbeing monitoring, location tracking, pet monitoring, and staff management in indoor communication [132]. The case study showed that LoRa technology is better suited for applications that tolerate delays and losses than dependent applications with stringent quality-of-service requirements. On the other hand, wearables like smartwatches require high data rates and low latency, which can be met by RedCap. RedCap can offer a peak bit rate of 50–150 Mbps with one to two weeks of battery life [34]. This allows RedCap to be used in wearable health monitoring and medical devices. Low-end augmented reality (AR), virtual reality (VR) glasses, and smart power grids are also able to utilize RedCap’s low latency and high data rate IoT services [22].

### 5.11. Work Safety

Many security policies exist for enhanced security for IIoT risk management and control security principles. Furthermore, the IIoT energy system can detect defects and energy usage of different components by continuous monitoring and real-time data processing. As a result, the system can prevent severe and harmful accidents and wasteful losses while also increasing overall energy efficiency. Monitoring and response systems can be well suited to long-range and low-power IoT protocols like LPWAN [133]. Porselvi et al. [134] implemented a low-powered smart alert system for the safety of mine workers using LoRaWAN. It is efficient in reducing death rates and disease by constantly observing the environment and alerting the workers about any potential danger. This system efficiently and economically obtains the mine worker’s medical data, which are then utilized for additional artificial-intelligence-based medical prognosis. On the other hand, new technology such as RedCap is suitable for CCTV video surveillance cameras with <500 ms latency with a 2–4 Mbps data rate, although high-end video, like smart farming, demands higher performance [34].

Table 3 summarizes the relevancy of the different IoT protocols from an application perspective to point out which protocol is suitable for certain usage criteria. In Table 4, we summarize some case studies conducted in countries in Asia, Australia, Europe, and South America to show practical usage of the protocols. The studies were conducted in both indoor and outdoor environments. In most cases, LoRaWAN was used to utilize IoT networks.

## 6. Open Issue

Although the aforementioned critical technologies lay the groundwork for future IoT connectivity, they only serve to demonstrate the qualitative viability of the aim. Moreover, many problems still need to be considered as IoT is a relatively young technology. In this section, we examine some of these issues for future investigation.

### 6.1. Scalability

The number of IoT devices needed to be deployed in the near future will be massive. And so, the scalability factor of an IoT environment is very important to keep in mind. The scalability of IoT systems can be considered from two perspectives [43]. One is load scalability, which refers to how scalable the traffic amount is for each device in a system. The other one is structural scalability, which refers to how scalable the system is in terms of the number of devices. For ISM band-centric protocols like Z-wave, Sigfox, and LoRa, interference becomes severe with denser environments because the frequencies are not reserved. LoRa uses a pure ALOHA channel access method, which is not very load-scalable due to high packet collision probability. Sigfox’s various restrictions, such as strict duty cycle, frequency hopping, and Listen Before Talk mechanisms, hinder its load scalability. Meanwhile, 3GPP licensed spectrum-based protocols such as NB-IoT, LoRa, and RedCap use OFDMA which is more load-scalable. From a structural scalability point of view, SigFox and LoRa are comparatively more device-scalable than their 3GPP counterparts, although Sigfox is the more scalable compared to LoRa [135]. LTE-M offers better device scalability compared to NB-IoT. Z-Wave’s scalability is comparatively low [136]. 3GPP-based protocols are much more load-scalable than ISM-based protocols due to their superior traffic handling capabilities, reserved dedicated spectrums, and fewer regulatory restrictions. Contrarily, unlicensed spectrum networks can accommodate a greater number of devices per area. Stakeholders may need to choose between these two sorts of scalability along with other desired IoT parameters during deployment. Use cases such as healthcare, smart buildings, vehicular communication, and meter readings can require high device density and, at times, high-traffic QoS. For tasks like these, NB-IoT and LTE-M can provide good load-scalable environments. On the other hand, LoRa and Sigfox will be able to accommodate the huge number of devices required in these scenarios. To support all these devices, existing LPWAN protocols may face certain challenges regarding interference, collision, massive data storage, and spectrum congestion, especially in the unlicensed spectrum. Cellular IoT using a centralized LTE model can support the present number of devices. However, soon, it will become a bottleneck with an increased number of devices. Also, duty cycle access mode in a large-scale network can result in packet collisions and retransmissions, resulting in spectrum congestion. Research needs to be conducted on service discovery among billions of devices connected to different nodes, decentralizing architecture using clustering approaches, building fog computing models, peer-to-peer communications, gateway densification, spreading factor assignment combined with joint scheduling, etc. [9,137,138,139].

### 6.2. Complexity and Interoperability

IoT systems are made of a vast number of heterogeneous devices like sensors and actuators that utilize various hardware and firmware. This makes even a single network quite complex. At present, IoT lacks defined architectural standards, which leads to difficulty in proper testing of the networks. Anyone can develop IoT technologies. On one hand, there are proprietary technologies like Sigfox and LoRa; on the other hand, there are 3GPP-developed protocols like NB-IoT and LTE-M. IoT devices running on one protocol need to be flexible enough to operate with devices running on other protocols in order to provide good QoS to the user’s overall task management. Otherwise, users may face vendor lock-in. And a network consisting of billions of devices is bound to be prone to security threats. Both the industry and academia are attempting to deal with this interoperability problem by standardization. Fog or edge computing can provide interoperability to some extent. Research needs to be conducted on edge computing to ensure efficiency and speed in computing, storage, and networking. Scalable interoperability solutions need to be designed because the network size will only increase, and interoperability needs to be maintained irrespective of the technology being used [140,141,142,143].

### 6.3. Integration

Integrating LPWAN protocols using unlicensed spectrum with cellular protocols can be rewarding but a very challenging task. The key performance indicators of these protocols differ from each other. Whereas cellular protocols provide better data rates, lower latency, and better QoS, the ISM band protocols can provide more range and less power consumption at a lower cost. Depending on the specific task, an IoT use case scenario can require all of these. In such a scenario, an LPWAN-5G integration can help users in many ways. However, this integration leads to several challenges. The hybrid architecture challenge occurs due to different mechanisms and different protocols used to communicate with the server. In order to make a successfully converging core network, the routing of different protocols needs to be carefully addressed. Cellular protocols use 3GPP security methods, whereas noncellular protocols’ security is often comparatively weak. Another challenge noncellular protocols suffer from is their weak ability to handle moving UEs, which can be addressed using a hybrid core utilizing both cellular and noncellular protocols. A hybrid core can be designed using Universal Mobile Telecommunications System Subscriber Identity Module (USIM) for LoRaWAN gateway, creating evolved packet data gateway (ePDG) in 5G core, incorporating LoraWAN gateway in cellular eNB, and vice versa. More research should be conducted to evaluate the performance of an overall scenario when simultaneously utilizing two or more protocols in the same system. To tackle security integration, network-independent security solutions need to be researched for LPWAN authentication. Currently, the 5G architecture mostly uses non-standalone architecture (NSA), which utilizes NR with the evolved packet core. With the upcoming 5G standalone architecture (SA), core hybridization can lead to a reduction in complexity and cost of deployment, operation, and maintenance. Also, a unified data management system will be required for such a scenario [37,144].

### 6.4. Security and Privacy

Due to its massive network size, security and privacy are both very challenging aspects to deal with. IoT networks face various security risks such as node tampering, signal jamming attacks, sleep deprivation attacks at the physical level, RFID spoofing, eavesdropping at the network level, encryption attacks like cryptoanalysis attacks, virus, DDOS attacks at the software level, and many more. Privacy threads to IoT networks include identification, profiling and tracking, lifecycle transactions, etc. These challenges need to be addressed in order to build a trustworthy IoT network because people’s lives and businesses will depend on many IoT use cases like smart homes, smart farms, and wearable health monitoring devices. Due to their being lightweight devices that are low in power consumption, complex security algorithms can become unfit for IoT devices. Also, the data regarding IoT will be of huge volume. Processing of this big data needs to be highly secured, otherwise, data breaches can happen. Research needs to be conducted on building tamper-resistant hardware, strong authentication methods, and regular firmware updates. Lightweight security systems need to be designed that are suited for IoT devices [142,145,146,147].

### 6.5. Single Point Gateway

Data are sent from sensors to LPWAN gateways. The gateways establish an IP connection to the Internet and send the data obtained from the embedded sensors to the Internet, which can be a network, a server, or a cloud. The gateways function as a transparent bridge, translating radio frequency (RF) transmissions to IP packets and vice versa, connecting to the network server via conventional IP connections. However, due to LPWAN technology’s single point of failure at the gateway section at the physical layer and a lack of redundancy, using gateways to communicate with end devices may become inefficient. Network, data, and application layer communication protocols can be more researched as they may avoid single-point failure [66].

### 6.6. ALOHA-Based Access

Deterministic traffic is gradually acquiring much significance in IoT environments. Advocates of Linux Open-source Hawaii Association-based (ALOHA) access need to be optimized for serving this type of traffic. Sadly, deterministic traffic handling is a typical restriction of the IoT MAC layer protocol, and this needs to be addressed. Research should be conducted on designing a hybrid or complete TDMA scheduler capable of allocating resources to ALOHA-centric access in addition to scheduling deterministic traffic [31].

### 6.7. Data Management

Data extraction, which can be viewed as gathering data from the appliances and extracting meaningful information from the obtained data, is an open issue that needs to be considered. Data extraction will significantly affect how well a system functions, notably when the number of appliances is expanded in a communication architecture. If the system has to be completely redesigned, it may be determined using the memory capacity, processing speed, and network bandwidth. In contrast to the problems with data extraction, data representation is a crucial area for research because it may facilitate information interchange between the IoT communication system and other technologies like ontology and semantic web technologies [148,149].

In Table 5, we present the difficulties that are prevalent in implementing an IoT network and the various challenges that come with it. The IoT network is relatively new, and many aspects of it are still being developed. In the table, some of the essential open issues that need to be researched and solved to make IIoT possible are scalability, complexity and interoperability, integration and combining protocols, single-point failures, ALOHA-based access, data management, security, and privacy. Apart from the abovementioned challenges, some other facts about IoT communication protocols must also be addressed. Numerous analysis studies of the protocols may consider the following topics: estimation of the collision rate, channel load, single device maximal throughput and maximum transmission unit (MTU), mobility and roaming, and proposing possible solutions for performance enhancement [150].

## 7. Conclusions

Due to scalability being one of the most important factors in its practical applications, IoT devices will be deployed in large numbers within a few years. However, so many architectures and protocols exist among IoT devices that it can easily become confusing as to which one is more suitable in an industrial environment and which one to avoid where. Some of the major technologies in the IoT market are Sigfox, LoRa, Z-Wave, NB-IoT, LTE-M, and RedCap. We presented an overview of the basic network architectures of these IoT communication protocols and a comparison that focuses on some of the key factors, such as low cost, long-range, and energy efficiency. The results of performance comparison and application perspective show that LPWAN-based protocols perform better and are more suitable than other IoT Protocols. Several parameters are essential to deciding which technology should be used in a certain industrial environment, which is covered in this study from an application perspective. However, there are some challenges in this relatively novel field of research that were also addressed in this paper.

## Figures and Tables

**Figure 1 sensors-24-02509-f001:**
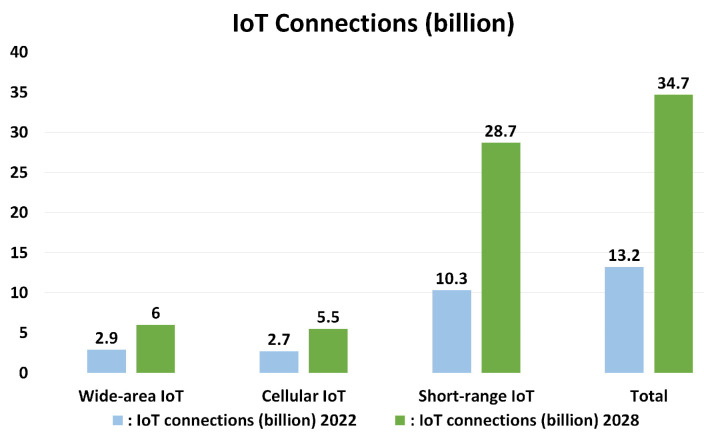
IoT connectivity number forecast (2022 and 2028).

**Figure 2 sensors-24-02509-f002:**
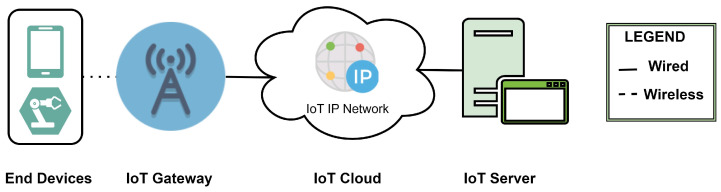
Network architecture for IoT protocols based on unlicensed ISM-band spectrum (Sigfox, LoRa, Z-Wave).

**Figure 3 sensors-24-02509-f003:**
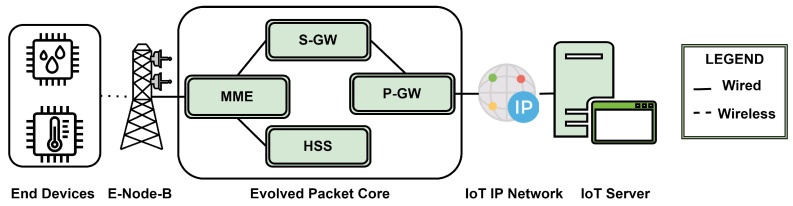
Network architecture for IoT protocols based on 3GPP licensed spectrum (NB-IoT and LTE-M).

**Figure 4 sensors-24-02509-f004:**
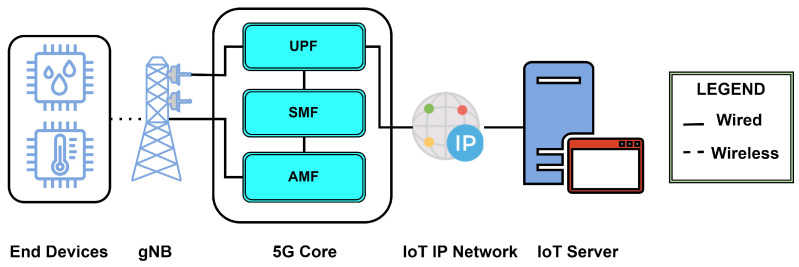
Network architecture for IoT protocol based on 3GPP licensed spectrum 5G NR based RedCap.

**Figure 5 sensors-24-02509-f005:**
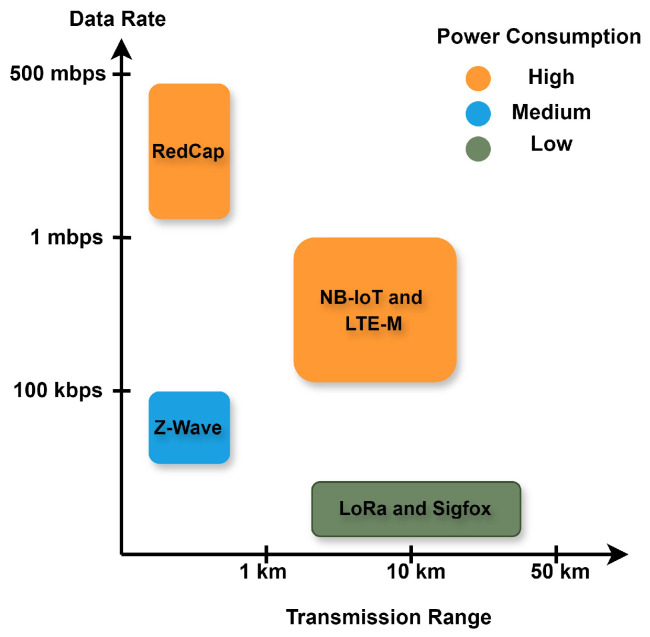
Performance comparison of different IoT communications protocols from coverage, data rate, and energy consumption points of view.

**Figure 6 sensors-24-02509-f006:**
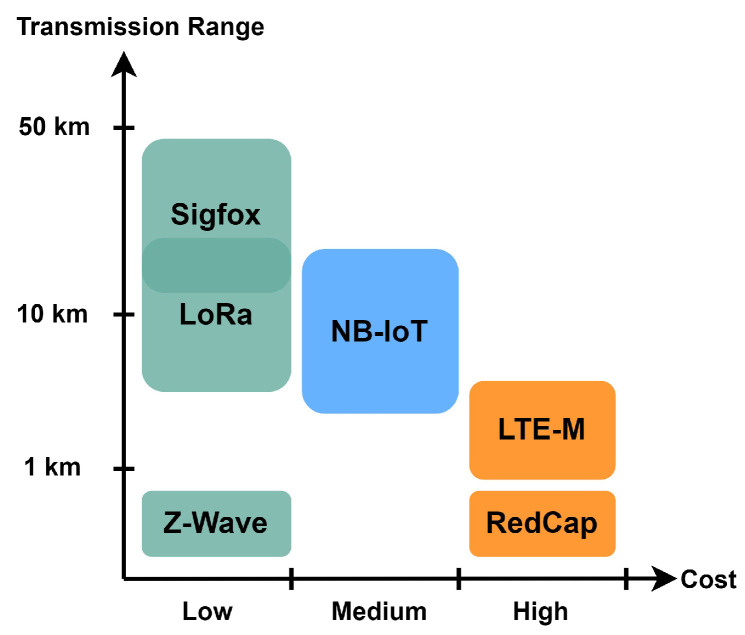
Performance comparison of different IoT communications protocols from coverage and cost-efficiency points of view.

**Figure 7 sensors-24-02509-f007:**
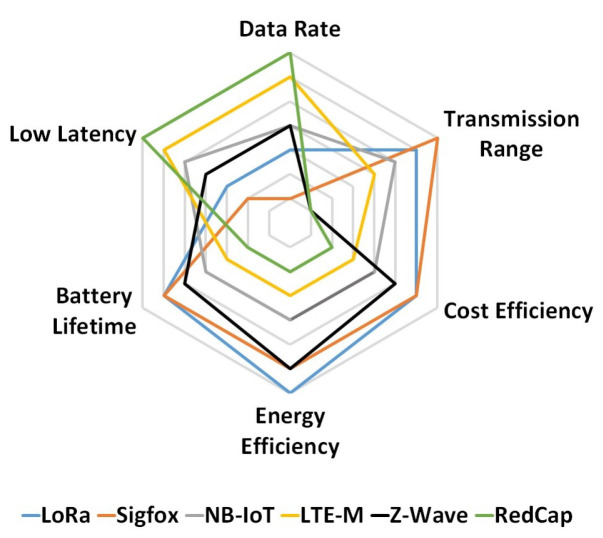
Performance comparison of different IoT communications protocols from data rate, transmission range, energy consumption, cost, and battery lifetime points of view.

**Figure 8 sensors-24-02509-f008:**
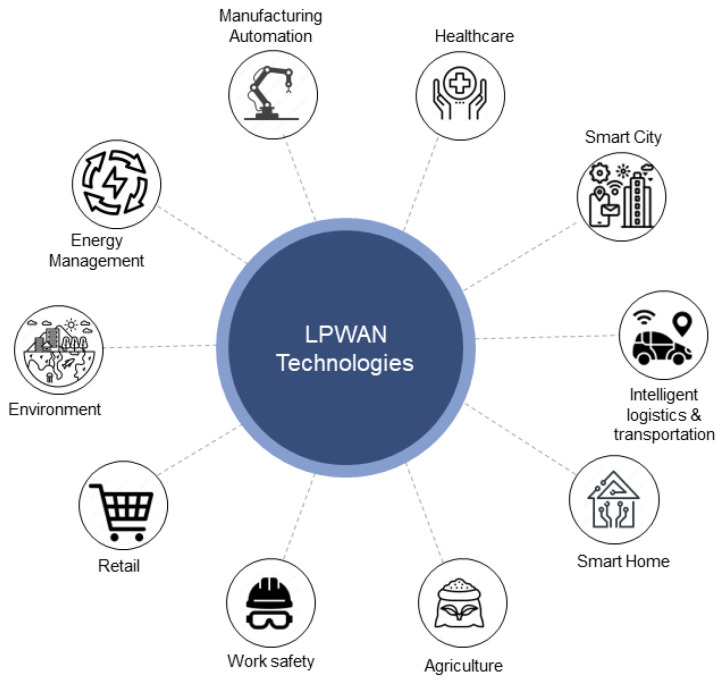
Applications of Industrial Internet of Things based on LPWAN protocols.

**Table 1 sensors-24-02509-t001:** Comparative analysis of existing literature of different IoT communication protocols.

Research	IoT Layer	Metrics	Protocols	Comparative Analysis	Experimental Analysis
Andre Gl’oria et al. [24]	Physical, Data link	Multinode capability, low-cost and power saving capabilities, delay, data rate	Wifi, Zigbee, LoRa, Bluetooth	✓	✓
Shadi Al-Sarawi et al. [25]	Physical, Data link	Power consumption, security, spreading data rate	6LoWPAN, ZigBee, BLE, Z-Wave, NFC, SigFox LPWAN	✓	✓
Ala’ Khalifeh et al. [23]	Physical, Data link	Small size, low cost, limited energy	LoRaWAN, Sigfox, NB-IoT and LTE-M	✓	✗
Burak H. Çorak et al. [36]	Application	Packet creation time, packet delivery speed, delay differences	CoAP, MQTT and XMPP	✓	✓
Thays Moraes et al. [26]	Application	Throughput, message size, packet loss	AMQP, CoAP and MQTT	✓	✓
JASENKA et al. [37]	Application	Latency, energy consumption and network throughput	MQTT, AMQP, XMPP, DDS, HTTP and CoAP	✓	✗
Kais Mekki et al. [29]	Physical	Network capacity, devices lifetime, cost, quality of service and latency	Sigfox, LoRaWAN and NB-IoT	✓	✗
H. Mroue et al. [27]	Physical	Carrier frequency, packet duration, number of channels and spectrum access.	LoRa, Sigfox and NB-IoT	✓	✓
Nadège Varsier et al. [34]	Physical	Bandwidth, coverage, data rate, latency reliability and battery life.	NB-IoT, LTE-M and RedCap	✓	✗
Teshager Hailemariam Moges et al. [35]	Physical	Bandwidth, range, data rate, battery life	NB-IoT, LTE-M, RedCap, EC-GSM-IoT	✓	✗
Our Research	Physical, Data link	QoS, security, power consumption, cost, coverage, datarate	LoRa, Sigfox, NB-IoT LTE-M, Z-Wave, RedCap	✓	✗

**Table 2 sensors-24-02509-t002:** Comparative analysis of different features of different IoT communication protocols.

Features	LoRa	Sigfox	NB-IoT	LTE-M	Z-Wave	RedCap
Standard	LoRaWAN	Collaboration with ETSI	3GPP	3GPP	Sigma Designs	3GPP
Frequency Band Type	Sub GHz ISM Bands	Sub GHz ISM Bands	Licensed Bands	Licensed Bands	ISM Bands	Licensed Bands
Frequency Band (GHz)	0.868 (EU) 0.915 (NA)	0.868 (EU) 0.915 (NA)	0.7, 0.8 0.9	Cellular Bands	0.868 (EU) 0.915 (NA)	0.4–7.1 (FR1) 24.2–52.6 (FR2)
Minimum Carrier Bandwidth (kHz)	125	0.1–0.6	200	1400	40–100	10,000
Data Rate (kbps)	50	0.1	250	Upto 1000	9.6–100	Upto 150,000
Transmission Range (km)	5–20	10–40	1–10	5	0.03	0.03
Energy Consumption	Very low	Low	Medium	Medium	Low	Medium
Cost	Low	Low	Medium	High	Low	High
Security	AES 128	Authentication and Encryption	AES 256	AES 256	AES 128 and CCM	-
Modulation	CSS	DBPSK, GFSK	QPSK	QPSK, 16-QAM and 64-QAM	GFSK	256-QAM, 64-QAM
Battery lifetime (Years)	>10	>10	>10	10	>10	<10
Link budget (db)	154	159	151	146	101	144
Reference	[20,29,60,71,80] [13,31,66,76,81]	[20,60,72,73,82] [13,29,66,77]	[19,32,60,61] [29,47,71,73]	[21,47,61,83] [29,35,71,73]	[62,84,85,86] [25,87,88,89]	[22,34,65] [35,57,59,90]

**Table 3 sensors-24-02509-t003:** Suitable IoT communications protocol based on application and physical features.

Use Cases	Features	Protocol
Smart City	Large coverage area, latency, cost	Sigfox and LoRaWAN [25,31,71]
Intelligent Logistics and Transportation	Low latency, high QoS, large coverage	NB-IoT [31,95]
Smart Farming and Agriculture	Large coverage area, latency, cost, low power	Sigfox and LoRaWAN [29,99,100]
Smart Home	Short range, lower latency, cost, low power	Sigfox and LoRaWAN [29,104]
Terminals for Retail Sales	Low latency, high QoS, large coverage	NB-IoT [29,106]
Smart Environment	Low latency, high QoS, large coverage	NB-IoT, LTE-M [111,113]
Smart Metering, Energy, and Grid	Long range, low power, robust QoS, and readability	NB-IoT [111,113]
Manufacturing and Automated Industries	Long range, low power, robust QoS, readability and cost	Sigfox and LoRaWAN and NB-IoT and RedCap [10,34,118]
Wearables and Health	Long range, low power, Robust QoS, readability and cost	Sigfox and LoRaWAN and NB-IoT and RedCap [22,34,50,131,132]
Work Safety	Low power, cost and readability	Sigfox and LoRaWAN and NB-IoT and RedCap [34,133]

**Table 4 sensors-24-02509-t004:** Summary of case study based on IIoT application area and IoT protocols.

Reference	Application Area	Geographical Location	Technology	Final Outcome
[94]	Smart City	Estonia	Sigfox, NB-IoT	In outdoor area both protocol provide coverage without delay, while in indoor NB-IoT perform better.
[96]	Intelligent Logistics and Transportation	Zhejiang Province, China	NB-IoT	To mitigate high power consumption and high deployment costs of wireless network, a smart parking system is developed.
[101]	Smart Farming and Agriculture	Italy	LoRaWAN	In terms of data transmission and energy efficiency performance observed in both indoor and outdoor area.
[105]	Smart Home	Bangkok, Thailand	LoRaWAN	Obtain the performance of LoRaWAN through a case study to explore communication ranges in both an outdoor and an indoor environment.
[127]	Asset tracking	Melville, Perth, Western Australia	LoRaWAN	Illustrated the suitability of using LoRaWAN to conduct temperature studies on local government assets in Australian metropolitan and residential areas.
[110]	Smart Environment	Brazil and Portugal	LoRaWAN	This study finds that LoRa performs incredibly well in crowded urban environments.
[112]	Smart Metering, Energy, and Grid	Lebanon	LoRaWAN	Suggests that LoRa can be used to build an open-source, inexpensive, modular system for energy metering applications.
[120]	Manufacturing and Automated Industries	Denmark	SigFox	Proposes a suitable deployment roadmap in smart manufacturing using LPWAN which is more suitable than Radio Frequency Identification (RFID).
[132]	Wearables and Health	Finland	LoRaWAN	Investigates the indoor performance of LoRa technology in remote health monitoring. The implementation shows that a small transmit power is enough to cover a large area.
[134]	Work Safety	India	LoRaWAN	Implemented a smart alert system for the safety of mineworkers that constantly observes the environment and alerts the workers. It is efficient in reducing death rate and disease.

**Table 5 sensors-24-02509-t005:** Future research agenda from insightful studies.

Theme	Challenges	Research Path
Scalability [9,43,135,136,137,138,139]	Spectrum congestion, packet collision, interference, becoming bottleneck	Clustering approaches, fog computing modes, peer-to-peer communications, gateway densification
Complexity and interoperability [140,141,142,143]	Lack of defined architectural standard, risk of vendor lock-in, security threads	Flexible protocol designing, standardization, fog or edge computing
Integration [37,144]	Different protocols use different mechanisms, data management, lack of good security in proprietary protocols	Designing a hybrid architecture, network-independent security solutions, unified database management
Security and Privacy [142,145,146,147]	Security threats like attack at physical level, network level, encryption, DDoS attack, privacy threats like authentication, identification, profiling	Building tamper resistant hardware, designing lightweight but strong authentication and encryption methods
Single-Point Gateway [66]	Single-point failure resulting in whole system failure	Network, data, application layer communication protocols rather than only physical level
ALOHA-based Access [31]	Restriction on deterministic traffic handling	Designing hybrid or complete TDMA scheduler
Data Management [148,149]	Data extraction will be a tough task in an ever expanding network	Redesigning systems based on memory constraints, processing speed, network bandwidth

## Data Availability

No new data were created or analyzed in this study. Data sharing is not applicable to this article.

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
