# Peer review of "Future Industrial Applications: Exploring LPWAN-Driven IoT Protocols"

_sensors, 2024, doi:10.3390/s24082509_

Round 1
Reviewer 1 Report (Previous Reviewer 4)
Comments and Suggestions for Authors
The paper notably improved after its revision, and I have no further comments. Well done!
Author Response
Thanks

Reviewer 2 Report (Previous Reviewer 3)
Comments and Suggestions for Authors
-
Expand on the comparative analysis: Consider providing more detailed comparisons between the LPWAN protocols (LoRa, Sigfox, NB-IoT, LTE-M) in terms of their performance metrics such as power consumption, coverage, data rate, security, cost, and QoS. This will enhance the reader's understanding and provide a more comprehensive basis for protocol selection.
-
Address scalability concerns: Discuss the scalability aspects of LPWAN protocols, highlighting how each protocol handles scalability challenges in industrial IoT deployments. Providing insights into scalability will help readers evaluate the long-term suitability of these protocols for evolving industrial needs.
-
Include real-world case studies: Incorporate real-world case studies or examples of industrial IoT applications using LPWAN protocols. This will illustrate the practical implications of protocol selection and provide readers with tangible examples to better understand the potential benefits and challenges in real-world scenarios.
Author Response
Please see attached

Reviewer 3 Report (New Reviewer)
Comments and Suggestions for Authors
In the manuscript, the authors have summarized the network architectures and compared the performance of the LPWAN-based IoT wireless protocols. They have also investigated some open issues at the end. The manuscript is well organized overall. However, some concerns still need to be addressed.
a) In Figure 1, would it be better to provide the number of IoT connections on the tops of the bars?
b) For the explanations of equations, there is no need to indent "where......".
c) Figures 1-4 are a little bit blurred. Could the authors please improve the quality of the presentation?
d) In Table 2, the authors provide a comparison among the IoT communication protocols. In terms of the frequency band, would it be better to provide some examples of the exact working frequency bands, e.g., 433MHz, 866MHz, 915MHz, etc. for LoRa?
Comments on the Quality of English LanguageIt would be better if the authors could further polish the writing.
Author Response
Please see attached

This manuscript is a resubmission of an earlier submission. The following is a list of the peer review reports and author responses from that submission.
Round 1
Reviewer 1 Report
Comments and Suggestions for Authors
The paper is a kind of survey/review of the most popular IoT wireless communications protocols, and as such it brings some useful information.
However, it is not well structured to meet the requirements for scientific paper, because it does not propose new concepts or interesting ideas in the field, as well as it does not demonstrate sufficient progress beyond the state of the art. The paper does not show precisely research design and results. Technological, scientific and methodological contributions are not convincingly presented.
The proposed work is more suitable to review paper, but even so it lacks detailed comparative analysis of the protocols in a quantitative term.
In line 432 there is missing number of Table ??
Comments on the Quality of English LanguageN/A
Reviewer 2 Report
Comments and Suggestions for Authors
If aurhors want to have a full review on LPWAN-Driven IoT Protocols. At this time, it is hard to avoid the 5G LPWAN standard, Redcap. Missing the latetst and the most popular 3GPP standard makes the review incomplete.
I also have concerns with table 2.
The transmission range of Z-Wave is only 3 meters?
The comparison table is useful as a review article, but it lacks the apple to apple comparison for example the battery life. The ~10 years of life are from different references which may not be related.
Reviewer 3 Report
Comments and Suggestions for Authors
-
Clarity and Definition: The paper asserts that IoT will lead Industry 4.0, but it would benefit from a more explicit definition of Industry 4.0 and its anticipated impact on various sectors.
-
Communication Complexity: Elaborate on the complexity of communication in IoT devices, addressing how it can impact device functionality and the overall effectiveness of Industry 4.0.
-
Protocol Diversity: While the paper mentions various communication protocols, it should delve deeper into how the diversity of these protocols might affect interoperability and standardization within the IoT ecosystem.
-
Wireless Overemphasis: Consider discussing the potential drawbacks or challenges associated with predominantly adopting wireless protocols, perhaps in terms of susceptibility to interference or spectrum congestion.
-
Critical Features Weighting: Provide insights into the weighting of the critical features in the comparative analysis. Are all features equally important, or does their significance vary depending on the industrial application?
-
LPWAN Suitability Criteria: The paper asserts LPWAN protocols' suitability for future industrial applications, but what specific criteria make them more suitable, and how were these criteria determined?
-
Application Perspective Depth: Expand on the IIoT application perspective, providing more detailed examples or case studies to illustrate how LPWAN protocols excel in real-world industrial scenarios.
-
Open Issues Significance: Discuss the significance and potential impact of the open issues highlighted, emphasizing why these need urgent attention and how their resolution can enhance the overall efficacy of IoT in industry.
-
Research Gap Identification: While the paper discusses open issues, it would be beneficial to identify any research gaps within the existing literature that this study aims to fill.
-
Alternative Solutions Consideration: Explore alternative solutions or protocols that might be overlooked in the current analysis, providing a more comprehensive overview of potential choices for industrial applications.
-
Temporal Consideration: Given the rapid evolution of technology, consider incorporating a temporal dimension to the study, discussing how the presented findings might adapt or change in the future IoT landscape.
-
Global Perspective: Consider addressing regional or global variations in the adoption and success of specific IoT protocols, as this could impact the generalizability of the study's findings.
-
Scalability Discussion: Discuss the scalability of the proposed LPWAN-based solutions, considering potential challenges or advantages as the number of connected devices in Industry 4.0 grows.
-
Ethical Implications: Touch upon the ethical considerations related to the adoption of IoT in industry, especially in terms of data security and privacy, and how different protocols may address these concerns.
-
Integration Challenges: Explore the challenges associated with integrating LPWAN protocols into existing industrial infrastructures, addressing potential disruptions or modifications required for a smooth transition.
Major revision.
Reviewer 4 Report
Comments and Suggestions for Authors
See attached file.

Comments on the Quality of English LanguageThe paper needs to be properly proofread.
Reviewer 5 Report
Comments and Suggestions for Authors
The article provides an overview of various physical layer communication protocols (Sigfox, LoRa, -Wave, NB-IoT, and LTE-M) in terms of power utilization, cost, range, data rate, QoS, and security.
1. While the article is easy to read, the reviewer did not find any novelty or significance.
2. Authors have made many claims at various places of the article without providing any reference or appropriate justification.
3. Authors have admitted that there are some comparative studies (summarized in Table 1) on IoT protocols in an experimental setting. They have also stated that IoT protocols have also been evaluated in simulated settings without an experimental setup (section 2 of the article). In this context, the article under consideration did not provide any simulation or experimental data to evaluate the target IoT protocols. In other words, the article does not contribute to the body of knowledge.
4. The data provided in Table 3 (Suitable protocol based on application and physical features) is oversimplifying the situation and therefore the provided recommendation for the protocols may be misleading.
5. IoT protocol parameters have been summarized in Table 2. This is the only point in the entire article where readers can find at least some information about the target protocols.
Comments on the Quality of English LanguageN/A